# Assessment of Thermal Osteonecrosis during Bone Drilling Using a Three-Dimensional Finite Element Model

**DOI:** 10.3390/bioengineering11060592

**Published:** 2024-06-10

**Authors:** Yung-Chuan Chen, Yi-Jung Tsai, Hao-Yuan Hsiao, Yen-Wei Chiu, You-Yao Hong, Yuan-Kun Tu, Chih-Kun Hsiao

**Affiliations:** 1Department of Vehicle Engineering, National Pingtung University of Science and Technology, Pingtung 912301, Taiwan; chuan@mail.npust.edu.tw (Y.-C.C.); edtuyk@gmail.com (Y.-Y.H.); 2Department of Medical Research, E-Da Hospital, I-Shou University, Kaohsiung 82445, Taiwan; ed108805@edah.org.tw (Y.-J.T.); d118030001@nsysu.edu.tw (Y.-W.C.); 3Institute of Medical Science and Technology, National Sun Yat-sen University, Kaohsiung 804201, Taiwan; d128030006@nsysu.edu.tw; 4Department of Orthopedics, E-Da Hospital, I-Shou University, Kaohsiung 82445, Taiwan; 5Department of Mechanical Engineering/Graduate Institute of Mechatronics Engineering, Cheng Shiu University, Kaohsiung 833301, Taiwan

**Keywords:** bone drilling, FE model, thermal bone necrosis, feed force

## Abstract

Bone drilling is a common procedure used to create pilot holes for inserting screws to secure implants for fracture fixation. However, this process can increase bone temperature and the excessive heat can lead to cell death and thermal osteonecrosis, potentially causing early fixation failure or complications. We applied a three-dimensional dynamic elastoplastic finite element model to evaluate the propagation and distribution of heat during bone drilling and assess the thermally affected zone (TAZ) that may lead to thermal necrosis. This model investigates the parameters influencing bone temperature during bone drilling, including drill diameter, rotational speed, feed force, and predrilled hole. The results indicate that our FE model is sufficiently accurate in predicting the temperature rise effect during bone drilling. The maximum temperature decreases exponentially with radial distance. When the feed forces are 40 and 60 N, the maximum temperature does not exceed 45 °C. However, with feed forces of 10 and 20 N, both the maximum temperatures exceed 45 °C within a radial distance of 0.2 mm, indicating a high-risk zone for potential thermal osteonecrosis. With the two-stage drilling procedure, where a 2.5 mm pilot hole is predrilled, the maximum temperature can be reduced by 14 °C. This suggests that higher feed force and rotational speed and/or using a two-stage drilling process could mitigate bone temperature elevation and reduce the risk of thermal osteonecrosis during bone drilling.

## 1. Introduction

In the surgical treatment of bone fractures, it is often necessary to predrill a pilot hole using a bone drill before inserting a bone screw for implanting. This facilitates the subsequent insertion along the pilot hole. However, the friction between the drill bit and the bone generates significant heat during bone drilling, resulting in thermal damage. The generated heat may lead to thermal damage to the bone tissues, known as bone necrosis or impairment of the bone-forming potential. It has been shown that temperatures over 50 °C are associated with irreversible changes in bone structure and physical properties [1,2]. The necrotic bone is resorbed through osteoclast activity, posing potential risks to the stability of bone screws and pins and ultimately leads to the failure of the fracture repair or implant fixation [3,4,5,6,7,8,9]. Consequently, some studies have focused on the rise in temperature during bone drilling and the associated necrosis phenomenon. While no clear consensus exists on the threshold or duration, the temperature in the cortical bone above 50 °C has been reported to decrease regenerative capacity and temperatures above 56 °C result in bone necrosis [3,4,6]. Research has demonstrated that an increase in temperature above 47 °C for one minute resulted in intense bone necrosis [3,10,11]. Ardan et al. [12] found that temperatures ranging from 43 to 68 °C in cortical bone delayed bone recovery. Moritz and Henriques [13,14] showed that bone tissue immediately became necrotic when the temperature surpassed 70 °C, with the severity comparable to that of 55 °C for 30 s. However, the majority of researchers believe that elevating the temperature above 47 °C may lead to thermal necrosis in human bone [3,11,15]. Consequently, the critical temperature associated with a high risk of osteonecrosis is commonly regarded as 47 °C in most current studies.

Many surgical drill designs focus on enhancing cutting performance and increasingly consider reducing the heat generated during operation to mitigate the complications from the rise in temperature effect during bone drilling. Clinically, factors influencing the temperature increase during bone drilling can be categorised into two main types: the design parameters of the drill bit and the usage parameters during the drilling procedure. The design parameters of the drill include the type of bone drill, diameter, point angle, helix angle, cutting edge, and the thinning between the cutting edge and chisel edge [16,17,18,19,20,21,22,23,24]. Davidson and James [16] explored the impact of bone drill geometric dimensions and bone thermal conductivity on bone temperature increases. Their results indicated that larger drill diameters or higher rotation speeds led to higher temperature rises, while larger helix angles or feed rates reduced temperature increases. Additionally, the results suggested that bone thermal conductivity has a greater impact on the temperature rise, while the point angle has less noticeable effects. The other type is the surgical parameters, including drilling speed, applied feed force, feed rate, bone quality, and cooling techniques. Some studies have investigated the effects of rotational speed, feed rate, and feed force on bone temperature [25,26,27,28,29,30,31,32,33,34]. However, consensus on the effects of drill diameter, drill speed, feed rate, feed force, and cooling technique on bone temperature remains to be determined and requires further investigation and confirmation.

Parametric studies can be efficiently conducted using finite element methods (FEMs), which not only reduce experimental costs but also enable the adjustment of various parametric conditions to identify optimal drilling parameters. By employing different analytical models, materials, and drill geometries, the prediction of expected outcomes is facilitated. Furthermore, simulation analysis allows for the exploration of parameters that are challenging to determine experimentally, such as bone friction, residual stresses, and thermal distribution. Given the substantial biological variability of animal and human bones, it is challenging to explore each parameter experimentally. Thus, adopting an experimentally validated FE model may effectively determine the optimal drilling parameters [35,36,37,38,39,40,41]. During the bone drilling process, the heat source is mainly concentrated at the tip of the bone drill, with the heat source changing with the drilling depth. Consequently, our study employed a three-dimensional dynamic elastoplastic finite element (FE) model to examine the thermal impact on bone. This approach enables us to investigate the thermal effect on the bone during and assess the thermally affected zone (TAZ) associated with bone heat. The TAZ means that this zone is at high risk of thermal osteonecrosis. To be conservative, in this study, the TAZ was identified as the region where the bone temperature exceeds 45 °C. Furthermore, we implemented a two-stage drilling approach, where a smaller hole was predrilled before the desired hole was drilled. This study explored the potential advantages of using this two-stage drilling approach in reducing bone temperature during drilling.

## 2. Finite Element (FE) Modelling

### 2.1. Temperature Distribution during Bone Drilling

The temperature field distributions in bone are governed by the conservation of heat. The temperature function (*T_bone_*) of the bone can be expressed by the three-dimensional heat transfer equation as follows [42]:(1)$$k_{b}[\frac{1}{r}\frac{\partial }{\partial r}(r\frac{\partial T_{bone}}{\partial r})+\frac{1}{r^{2}}\frac{\partial^{2}T_{bone}}{\partial \theta^{2}}+\frac{\partial^{2}T_{bone}}{\partial z^{2}}]=\rho_{b}c_{b}\frac{\partial T_{bone}}{\partial t}$$
where $k_{b}$, $\rho_{b}$, and $c_{b}$ are thermal conductivity, bone density, and specific heat of bone, respectively. *r*, *θ*, and z express the cylindrical coordinates.

Thermo-mechanical equations have been developed from machining theory to predict the heat generated during drilling. The amount of heat generated when the bit is in frictional contact with the bone can be considered as the heat generation rate [43]:(2)$$G=\dot{q_{f}}+{\dot{q}}_{p}$$
where ${\dot{q}}_{f}$ is the heat generated by the friction between the bit and bone in N-m/s, and ${\dot{q}}_{p}$ is the heat generated by the deformation of the plastic during cutting (N-m/s), which can be expressed as [44]:(3)$${\dot{q}}_{p}=\eta \sigma_{e}{\dot{\varepsilon }}^{pl}$$
(4)$${\dot{q}}_{f}=2\pi Rn\mu F_{n}$$
where η is the inelastic heat fraction, $\sigma_{e}$ is the effective stress (MPa), ${\dot{\varepsilon }}^{pl}$ is the plastic strain rate (1/s), R is the drill radius (mm), n is the rotational speed (rpm), μ is the friction coefficient, and $F_{n}$ is the normal force applied during drilling (mm). The heat flux generated by the friction between the bone and drill bit can be expressed as [43]:(5)$$q_{f}^{\prime \prime }=q_{bone}^{\prime \prime }+q_{drill}^{\prime \prime }$$
where $q_{bone}^{\prime \prime }$ is the heat flux transmitted to the bones in W/m^2^, and $q_{drill}^{\prime \prime }$ is the heat flux transmitted to the drill bit in W/m^2^. It is assumed that the entire frictional power will be converted into heat and transferred to the drill bit and bones. If the heat partition factor is $\eta_{1}$, then the heat flux transferred to the drill bit and bone is expressed as [44]:(6)$$q_{drill}^{\prime \prime }=\eta_{1}q_{f}^{\prime \prime }$$
(7)$$q_{bone}^{\prime \prime }=(1-\eta_{1})q_{f}^{\prime \prime }$$

### 2.2. Construction of FE Model

During the bone drilling process, the bone temperature changes over time. The analyses conducted in this study consider the dynamic interplay among friction, temperature, and stress fields. Therefore, a three-dimensional (3-D) dynamic elastoplastic finite element model was developed to simulate the temperature rise in bone during the drilling process. As the temperature of the bone varies with time throughout the drilling procedure, a dynamic temperature–displacement analysis module was utilised. This module is based on the explicit integral operating principle provided by ABAQUS (2023) software (Dassault Systemes, ABAQUS Inc., Waltham, MA, USA), offering a precise and efficient technique for solving temperature field complexities in contact problems. In the model, the temperature distribution of the bone during drilling was assumed to be localised around the drill hole, and a cylindrical shape (with a thickness of 4 mm and a diameter of 8 mm) was utilised to simulate the bone. The degrees of freedom in all six directions are constrained along the outer peripheral surface of the bone cylinder. The drill bit diameters are 3.5 mm and 4.5 mm, with point angles of 108 degrees and helix angles of 23 degrees. We used two rotational speeds (*n* = 800 and 2000 rpm) and applied four different axial forces (10 N, 20 N, 40 N, and 60 N) at the top of the drill bit to simulate feed forces. These settings were chosen based on clinical experience and the common screw sizes used for fracture fixation. The drill bit is made of high-quality 316L stainless steel, and the selected feed forces are consistent with previous literature [6,7,45]. The cortical bone thickness was set at 4 mm, and rotational speeds were set at 800 and 1200 rpm. The origin of the x–z coordinates was defined at the edge of the drill hole diameter, with the z-coordinate indicating the drilling depth. Eight-node hexahedron elements were utilized for meshing both the bone and the drill bit. Figure 1 illustrates the geometry and mesh models of the drill bit and cortical bone.

In our FE model, the frictional contact surface between the bone and the drill was set as a contact pair. The harder surface is designated as the master surface, while the softer surface is identified as the slave surface. In our model, the contact surface of the drillwas assigned as the master surface, and the contact surface of the bone was specified as the slave surface. The frictional contact between the drill bit and the bone was characterised as face-to-face contact. The definition of face-to-face contact is that, when two objects are in contact, surface-to-surface contact discretisation is used in the interactive calculation during the contact process to avoid erroneous contact due to the intrusion of the active surface into the passive surface. The heat generated in the contact area flows into the bone and drill bit. A heat partition factor *f* weights the heat flux into the slave surface, while the heat flux into the master surface is weighted by (1-*f*). In this study, it is assumed that heat is initially evenly distributed between the two surfaces. The analytical model assigns an initial value of 0.5 as the heat partition factor.

Ductile damage criteria were employed to model the damage evolution and predict failure. In this study, one criterion specifies that damage initiation occurs at a ductile failure strain of 0.008 [31]. The criterion for element removal is based on an effective plastic displacement value of 0.3 mm, at which point bone elements are removed. This value was determined from the chip morphology of bone drilling [32]. This damage criterion has been successfully applied in finite element bone drilling models and has been used to control the removal of elements during the drilling procedure [39,40]. During drilling, the position of the contact point changes continuously on the bone and the drill bit surfaces. Consequently, after a short drilling period, the cold bone absorbs more heat, and comparatively less heat flows into the drill bit. Therefore, the heat partition factor is not constant and varies with the drilling time. In the simulations, the thermal contact behaviour between the drill bit and bone was modelled using surface-to-surface contact discretisation. In addition, the contact interaction properties must be defined for the contact pair. An assumption is made that Coulomb’s friction law governs the friction behaviour between the drill bit and bone: the coefficient of friction is assumed to be 0.3 [9]. The thermal conductivity for cortical bone and drill bit are 0.452 and 16.2 (W/m·k), respectively [39,40].

### 2.3. Material Properties and Boundary Conditions

The drilling model employed in this study involves two primary materials: bone and the drill bit. The drill bit is constructed from 316 stainless steel (type 310.25 and 310.31 for diameters of 2.5 and 3.5 mm, respectively), and the mechanical properties of the materials are detailed in Table 1. Feed forces of 10, 20, 40, and 60 N were applied to the centre of the drill bit. The measured room temperatures (22 and 25.5 °C) were input as the initial bone temperature for analysis. In the FE model, we assumed that almost all of the energy used in material removal is converted into heat. Bone is a poor conductor of heat; the external surface of the cortical bone is less affected by the heat flow during hole drilling, and the temperature is the same as the surrounding environment. Thus, the external surfaces of the cortical bone are considered adiabatic boundary conditions, i.e., the effects of environmental heat convection and radiation are not taken into account [12,13,14,15,16].

## 3. Bone Drilling Experiment

An in vitro bone drilling test was conducted under controlled laboratory conditions to validate the analytical results of the FE model. A bone drilling platform was designed to install the measurement instrumentation. It was equipped with an infrared positioning system to determine the drilling position. The experimental platform, consisting of a personal computer (PC), electronic data acquisition system (FLUKE 2860A, WA, USA. Frequency: 0.01 s), thermocouple (K-type, Ni-Cr/Ni-Al, MTI Corp. Accuracy: ±0.5 °C), torque sensor (RT25, UAS. Accuracy: 0.01%), load cell (AL50, Japan, Accuracy: 0.2%), DC motor controller (9B060S-2N, TROY, Shihlin, Taipei, Taiwan. Accuracy: ±5 rpm), fixture (jig), linear guide (slide), and pulley system, was designed to carry out the bone drilling test. The rotational speed was controlled by the DC motor, and a weight (1, 2, 4, or 6 kg) was used to provide the constant feed force (10, 20, 40, or 60 N) through the guide and pulley system. During bone drilling, the experimental signals were transmitted and stored in the data acquisition system. The experimental platform for bone drilling is depicted in Figure 2a.

A fresh porcine femur with a cortex thickness of over 5 mm was selected for the experiment. Porcine bone was chosen because its material properties closely resemble those of human bone [46,47]. The porcine bone was dried for 1 h to avoid any possible influence on the measured temperature of the cooling effect of the water on the bone surface. Then, the cortical bone was cut into a 20 mm × 20 mm specimen, clamped onto the fixture, and installed on the experimental platform. For bone temperature measurement, four thermocouples were embedded into the drill hole at an angle of 90° and at a distance of 0.5 mm from the edge of the drilling hole with depths of 2 mm and 4 mm from the bone surface (Figure 2b).

To validate the FE model, experiments were performed to measure bone temperature. During the experiments, the measured environmental temperatures 22.0 °C and 25.5 °C and the relative humidity (78%RH) corresponded to the initial temperatures and humidity of the bone specimens. As drilling progressed, the bone temperature changed, and the thermocouple signals were continuously transmitted to a computer system through an electronic data acquisition system. The electronic data collector can simultaneously capture temperature data from 20 thermocouples with a maximum frequency of 0.01 s. Upon completion of drilling, the drill bit was retracted to its original position, its rotation was stopped, and it was allowed to cool naturally. In the validation of the FE model, the measured temperatures, 22.0 °C and 25.5 °C, were used as the initial bone temperatures for the drilling process. Thermal paste (silicon) was used to fill the thermocouple holes, minimizing heat loss from air gaps. During the bone drilling test, the temperature signals from the thermocouples were transmitted to the computer via the temperature data acquisition card (NI 9216) and stored for analysis. Feed forces of 10, 20, 40, and 60 N were applied using the weight and pulley system, and drilling was performed to a depth of 5 mm. Bone temperatures corresponding to rotational speeds of 800 and 1200 rpm were recorded and compared with FEM results. The results of this experiment can be used to validate the finite element model, confirming the accuracy of the simulation results.

## 4. Results

### 4.1. Validation of the FE Model

The experimental data were used to compare and validate the accuracy of the FE model’s analytical results. Figure 3 illustrates a comparison between the analytical results from the FE model and the experimental data (mean, maximum, and minimum temperature) which were obtained from 30 boreholes. In Figure 3a, using a drill bit diameter of D = 3.5 mm, a feed force of F = 20 N, a rotational speed of *n* = 1200 rpm, and a bone density of 1.64 g/cm³, measurements were taken at positions x = 0.5 mm from the drill hole edge at a depth of 4 mm. The results showed peak bone temperatures of 33.6 °C and 34.6 °C for the experiment and simulation, respectively. The corresponding drilling times at these peak temperatures were 6.7 s and 7.4 s, respectively. The maximum differences observed in peak bone temperature and corresponding drilling time were 1.0 °C and 0.7 s, respectively. Figure 3b displays the temperature–drilling time curves for both FE simulations and experiments. It was observed that the maximum bone temperatures in the experiment and simulation were 33.5 °C and 34.0 °C, respectively. Correspondingly, the drilling times were 6.9 s and 7.5 s, respectively. The variances in the maximum bone temperature and the corresponding drilling time were 0.5 °C and 0.6 s, respectively. Based on these findings, the FE model is considered sufficiently accurate for predicting the impact of the temperature increase during bone drilling.

### 4.2. Maximum Temperature with Radial Distance

Figure 4a–d shows that the curves of the maximum temperature at different depths (z = 1 to 4 mm) decrease as the radial distance increases when drilling with two bone drill diameters (D = 3.5 and 4.5 mm) and two rotational speeds (N = 800 and 2000 rpm). From the curves shown in Figure 4a,b, it can be observed that when the drill diameter is 3.5 mm, temperatures above 45 °C are reached within a radial distance of less than 0.2 mm; hence, the region prone to thermal osteonecrosis is approximately within this 0.2 mm. However, when drilling with a 4.5 mm diameter bone drill, the curves in Figure 4c,d show a rotational speed of 800 rpm, and the thermally affected zone (TAZ) extends to approximately 1.0 mm. In contrast, when the speed increases to 2000 rpm, the TAZ reduces to below 0.4 mm. This indicates that higher rotational speeds can reduce the thermal osteonecrosis risk region. However, a greater drill bit diameter may increase the TAZ.

### 4.3. Effects of Feeding Force on Bone Temperature

Figure 5a shows the temperature variation over time at a distance of 0.1 mm from the hole at a depth of 1 mm when using a 3.5 mm diameter bone drill at a speed of 800 rpm with applied thrusts of 10, 20, 40, and 60 N. The curves show that when a 60 N feed force is applied, the maximum temperature is lower compared to feed forces of 10 N, 20 N, and 40 N, and the time to reach the highest temperature is the shortest (approximately 1 s). Conversely, when the feed force F is 10 N, the temperature is highest, and the time to reach the highest temperature is the longest (approximately 9 s). A higher feed force can reduce the temperature during the drilling process. Figure 5b illustrates the relationship between the maximum temperature and the radial distance during drilling. The graph demonstrates an exponential decrease in the maximum temperature with radial distance. When the feed forces are 40 and 60 N, the maximum temperature does not exceed 45 °C. However, when the feed forces are 10 and 20 N, the maximum temperature exceeds 45 °C within a radial distance of 0.2 mm, indicating a high-risk zone for potential osteonecrosis.

### 4.4. Effects of the Predrilled Hole on Bone Temperature

In this study, a two-stage drilling procedure was used to investigate the effect of drilling on bone temperature. The procedure involved initially drilling to the desired depth with a smaller diameter drill bit, followed by drilling with a specified larger diameter drill bit. In the analytical model, predrilling was performed using a 2.5 mm diameter drill bit. This was followed by drilling with the specified bone drilling diameters of 3.5 mm or 4.5 mm to predict the effect of the second drilling stage on temperature increase. Figure 6a depicts the temperature-time curve using a 3.5 mm bone drill with a rotation speed of 800 rpm and a feed force of 20 N. In Figure 6a, two curves represent the temperature response during bone drilling in two different drilling procedures: One shows the temperature curve during bone drilling using a 3.5 mm drill bit directly (without predrilling), and the other involves making a smaller predrilled hole (Dp = 2.5 mm) before the 3.5 mm bone drilling (with predrilled, two-stage drilling). Analysis of the results revealed that at the edge of the drilling site 0.1 mm (x = 0.1 mm) from the surface and at a depth of 1 mm (z = 1 mm), the maximum temperature during drilling with the bone drill is 56 °C. However, when a 2.5 mm pilot hole is predrilled before bone drilling with the 3.5 mm bone drill, the maximum temperature reduces to 42 °C, resulting in a 14 °C decrease in bone temperature. This suggests that predrilling a smaller hole before the bone drilling can effectively reduce the temperature during bone drilling. Figure 5b illustrates the bone temperature rise curve at different depths when predrilled with Dp = 2.5 mm before D = 3.5 mm bone drilling. The graph shows that the bone temperature is less than 47 °C at various depths. This result indicates that predrilling can effectively reduce bone temperature rise and mitigate the risk of thermal bone necrosis.

### 4.5. Effects of Rotational Speed on Thermally Affecting Zone (TAZ)

Figure 7a depicts the variation in the TAZ with drilling depth at rotation speeds of 800 rpm and 2000 rpm using a 3.5 mm bone drill diameter. Regression analysis indicates that the TAZ area increases linearly with drilling depth at both speeds, with the slower speed resulting in larger TAZ areas than the higher speed. At a rotational speed of 800 rpm, increasing the drilling depth from 1.0 to 4.0 mm increased the TAZ by about 2.7 times (from 0.3 to 0.8 mm). However, at 2000 rpm, the TAZ increased by about 3.5 times (from 0.15 to 0.53 mm).

Figure 7b shows the variation in the TAZ area with the drilling depth at rotation speeds of 800 rpm and 2000 rpm using a 4.5 mm bone drill diameter. Similar trends are observed under both speed conditions, with the TAZ area increasing linearly with drilling depth. Again, the slower speed resulted in larger TAZ areas than the higher speeds. At a depth of z = 4.0 mm, the TAZ for 800 and 2000 rpm are 2.2 mm and 1.3 mm, respectively. At a rotational speed of 800 rpm, increasing the drilling depth from 1.0 to 4.0 mm increased the TAZ by about 2.5 times (from 0.5 to 1.3 mm). However, at 2000 rpm, the TAZ increased by about 2.2 times (from 1.0 to 2.2 mm).

## 5. Discussion

In this study, we employed a dynamic elastoplastic FE model to analyse the impact of variations in drilling speed, feed rate, and feed force on the maximum bone temperature generated at the drilling site during bone drilling. Our goal was to evaluate the potential risk of these parameters in inducing bone necrosis. The rationale for utilising the simulation analysis is that it enables the effects of each parameter and their interactions on the heat generated during the bone drilling process to be differentiated within the same study. Some experimental studies using thermocouples to measure bone temperature, often placed near the drilled hole’s edge, only provide temperature readings near the drill hole rather than the actual bone temperature at the drilling site. In contrast, finite element analysis (FEA) allows for the temperature to be calculated at any location alongside the assessment of the gradient and range of the temperature distribution within the bone. This capability facilitates evaluating the high-risk range for thermal necrosis due to temperature elevation.

While some minor discrepancies exist between our model and the experimental results, the difference could be due to material properties and environmental differences. The moisture or blood cooling in human bone could prevent heat from being generated during bone drilling. Our analytical findings from the model demonstrate comparable accuracy to the experimental data in predicting temperature increases during bone drilling. Both sets of results were obtained under in vitro conditions, devoid of blood flow; thus, potential variations may arise when applied to clinical scenarios. Heat transferred to living tissue can lead to thermal damage and cell apoptosis. Studies indicate that heating bone above 50 °C can result in irreversible changes to its physical properties. Thermal bone necrosis and delayed healing have been observed in canine bone [1,2,3,4]. While our study identified the TAZ conservatively as bone temperatures exceeding 45 °C, we believe efforts should be made to minimise thermal injury to a level considered safe for local osteoblasts.

Matthews et al. [6] conducted experiments using human cortical bone, where they observed the highest temperature at 0.5 mm from the hole edge during drilling. They applied feeding forces of 2, 6, and 12 kg and rotation speeds of 345, 885, and 2900 rpm. Their results showed that both feed force and rotation speed affected the increase in bone temperature, with higher feed forces and rotation speeds effectively reducing the temperature rise. Inan et al. [48] investigated the temperature increase during bovine bone drilling, examining the effects of different rotation speeds (600, 900, 1200, and 1800 rpm). The results showed that higher rotation speeds resulted in a smaller necrotic impact zone. At a rotation speed of 1200 rpm, the highest temperature reached was 48.4 °C, while at a lower rotation speed of 600 rpm, the temperature rose to as high as 152 °C. Mustafa et al. [49] conducted experiments using bovine cortical bone with a diameter of 0.27 mm. They explored the effects of different forces (2, 3.8, 4.8, and 6.2 N) and high rotation speeds (20,000–100,000 rpm) on the rise in temperature in bovine bone. Their results also indicated that increasing force or rotation speed shortens the friction time, reducing heat generation.

Bachus et al. [45] examined the impact of various forces, specifically 53, 83, 93, and 130 N, on temperature rise at a constant rotational speed of 820 rpm, with temperature measurements taken 0.5 mm from the drill hole. Their findings indicate that, as drilling forces escalate from 57 to 130 N, both the maximum temperatures and their duration above 50 °C can be effectively reduced. This reduction potentially lowers the occurrence of thermal necrosis in the surrounding cortical bone. Our results imply that increasing the feed force promotes the effectiveness of bone cutting and reduces the drilling time, effectively reducing the rise in bone temperature caused by drilling. In this study, the parameters were set using clinical applications, and the results of our analyses show a similar trend in the effect of this parameter on temperature as observed in the current literature.

When comparing the results in Figure 6a,b, it is found that the TAZ range, when drilling with a smaller drilling diameter, is smaller than when using a larger drilling diameter. This has been confirmed by our previous study [39,40]. This is because drilling larger diameter holes facilitates an increased path for heat flow into the bone, leading to a more pronounced rise in bone temperature. Furthermore, both figures demonstrate a consistent trend: When drilling at higher speeds, the TAZ area is notably reduced compared to lower speeds. This phenomenon is attributed to the heightened cutting efficiency associated with increased rotational speeds, facilitating quicker completion of the drilling process and, consequently, reducing the duration of the heat transfer during drilling, thereby mitigating heat generation. The main source of heat during drilling is the frictional heat generated by the cutting between the drill tip and the bone. Increasing both the spinal speed and the feed force allows the drill to cut more efficiently than at slower speeds, thus generating less frictional heat. Clinically, the affected zone of thermal damage is difficult to measure experimentally; the FE model provides a reliable approach to assess the potential risk region of thermal damage.

An ideal experiment for determining the bone temperature during drilling is difficult because real bone is a complex and anisotropic biological tissue. The development of 3D models of real machining operations in surgery, including anisotropic constitutive modelling, is one of the challenges because the accuracy of the simulated model depends greatly on the thermo-mechanical properties and the constitutive behaviour of the bone tissue. The anisotropic and heterogeneous structure of the cortical bone could be approximated as an isotropic homogeneous material due to the relatively low mechanical anisotropy of the cortical bone tissue [50].

While numerous technical studies have explored methods to mitigate the heat effects on bone, such as utilising coolant irrigation, optimising drill bit geometry [9], enhancing wear resistance [51], implementing step-power drilling [52], etc., there is no conclusive evidence to indicate that irrigation cooling been universally adopted as a standard practice, despite strong recommendations from major implant manufacturers. However, clinically, irrigation cooling is standard practice and is strongly advocated for by leading implant manufacturers [53,54,55,56]. Although coolant irrigation applied to the bone surface can effectively lower surface temperatures, insufficient evidence confirms its penetration into the borehole for temperature reduction. Most of the current literature has focused on the temperature field in single-pass drilling and the maximum temperature adjacent to the drill tip. This study conducted a two-stage process to reduce the temperature during bone drilling. Firstly, a predrilled hole with a smaller diameter was drilled, and then the designed bone drilling was performed following the direction of the predrilled hole. This two-stage drilling process could significantly reduce the cutting area between the drill tip and the bone after the predrilling stage, thereby substantially minimising frictional heat during the subsequent drilling stage. While this two-stage drilling process may add complexity compared to a single-stage one, our study shows that it may be considered an effective method for mitigating bone temperature elevation during drilling; however, more experimental and clinical studies are needed to validate the approach.

This study has several limitations. Firstly, the FE model does not replicate an exact clinical situation, given that surgery is manually operated and bone properties vary widely. Blood and tissue humidity were not considered in the model. Secondly, the current FE model did not consider the irrigation and/or cooling systems. Thirdly, it is difficult to find appropriate surgical parameters or material properties for the model because of the inherent variability in human tissues. Fourthly, the level of thermal bone necrosis identified is based on the limited data in the literature, meaning the estimated thermal damage remains unjustified; more clinical or experimental data are needed to confirm the thermal damage dose. Lastly, in the two-stage drilling process, the model did not account for the heat generated during the first stage, which could potentially interact with the second stage. This interaction could lead to an increase in drilling temperature during the second stage. However, further research or experiments are needed to confirm this effect.

## 6. Conclusions

This study utilised a three-dimensional FE model to evaluate bone temperature rise during the bone drilling process and assess the high-risk zone for potential thermal bone necrosis. The investigated parameters included drill bit diameter, feed force, and three rotation speeds. In summary, the following conclusions are drawn:

Our three-dimensional FE model was experimentally validated and can effectively assess temperature elevation and the thermally affected zone (TAZ) during bone drilling. This is valuable for identifying optimal drilling parameters, designing drill handpieces, and implementing robot-assisted bone drilling.Higher rotational speeds may reduce bone temperature and decrease the TAZ; however, the TAZ increases with drill depth, feed force, and drill bit diameter.Implementing a two-stage drilling process can minimise frictional heat, thereby reducing temperature during bone drilling. This approach may be considered an effective method for mitigating bone temperature elevation during drilling.

## Figures and Tables

**Figure 1 bioengineering-11-00592-f001:**
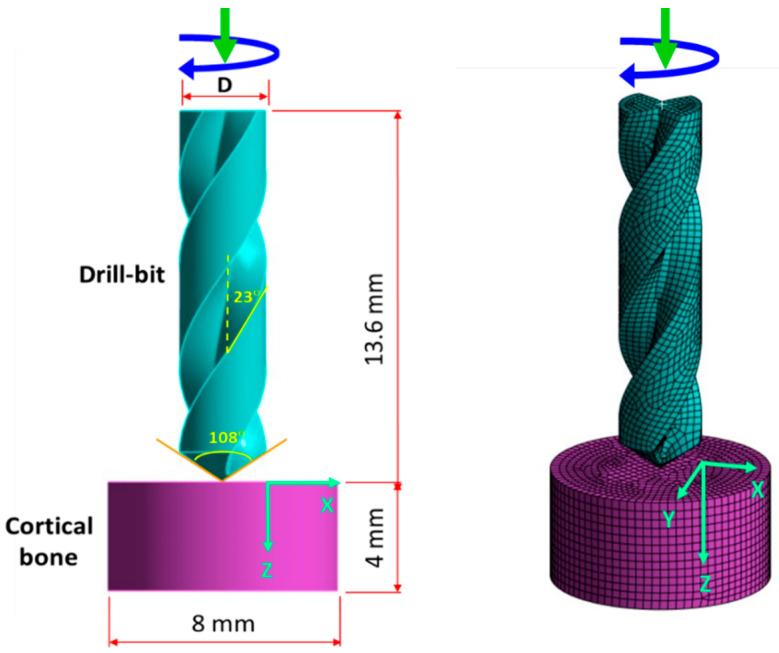
Geometry and mech models of the drill-bit and cortical bone.

**Figure 2 bioengineering-11-00592-f002:**
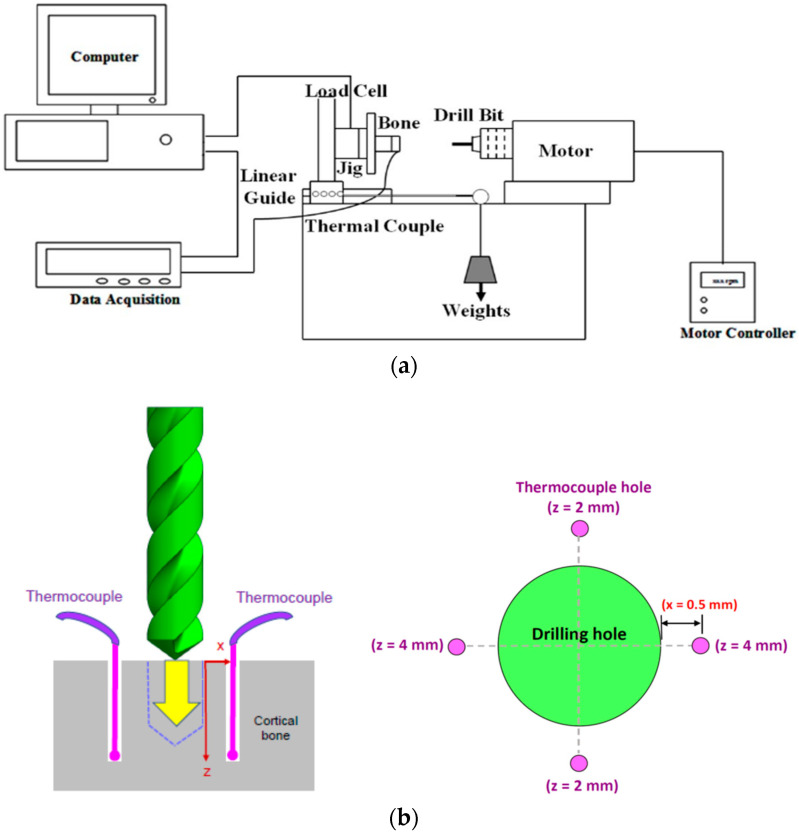
(**a**) Experimental setup for bone temperature measurement and (**b**) schematic illustration of the embedded positions of thermocouples at a distance of 0.5 mm from the edge of the drilling hole with depths of 2 mm and 4 mm from the bone surface (x = 0.5 mm; z = 2 mm and 4 mm).

**Figure 3 bioengineering-11-00592-f003:**
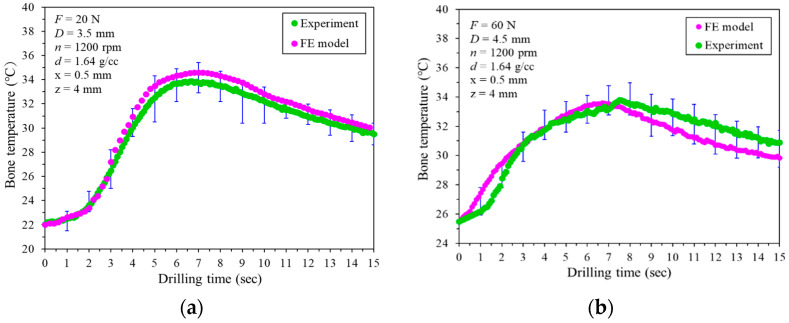
Comparisons of the bone temperature from the FE model and the experiments at the position x = 0.5 mm and z = 4 mm. The rotational speed is n = 1200 rpm and the bone density is d = 1.6 g/cc. The conditions are as follows: (**a**) feed force = 20 N, drill bit diameter D = 3.5 mm; (**b**) feed force = 60 N, drill bit diameter D = 4.5 mm. The upper and lower whiskers (⊤ and ⊥) display the experimental (green) maximum and minimum temperatures from 30 drilling boreholes, respectively.

**Figure 4 bioengineering-11-00592-f004:**
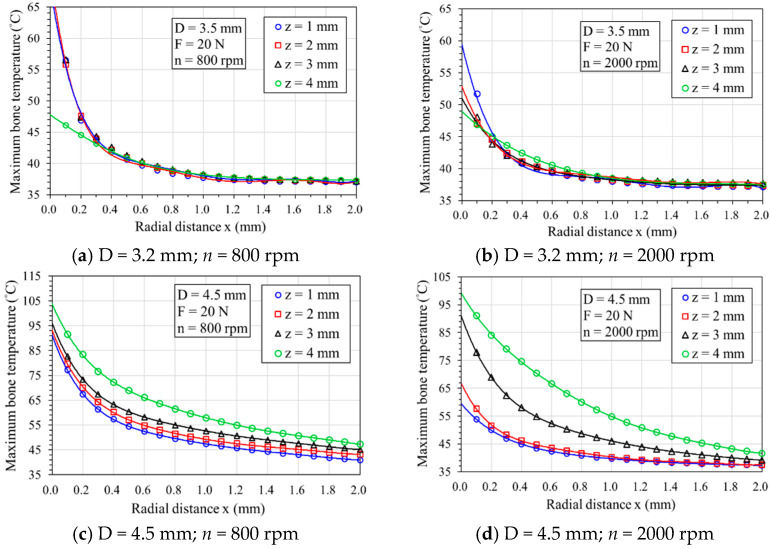
Maximum temperature with radial distance at different depths (z-direction).

**Figure 5 bioengineering-11-00592-f005:**
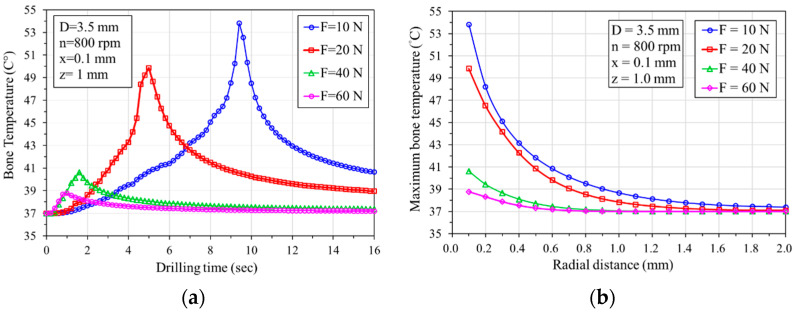
Effect of feeding forces on bone temperature: (**a**) temperature–drilling time relationship curve; (**b**) maximum temperature decreases exponentially with radial distance.

**Figure 6 bioengineering-11-00592-f006:**
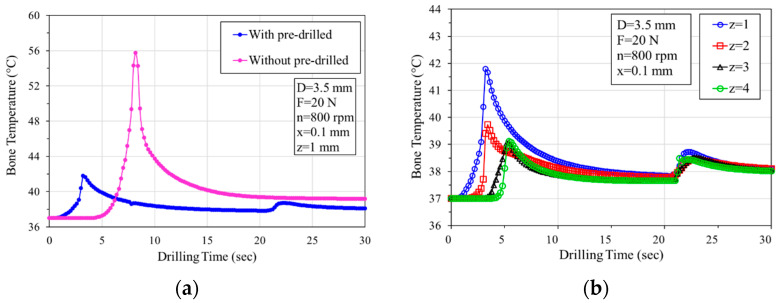
Bone temperature–drilling time responses: (**a**) with/without predrilling (predrilled holes of 2.5 mm diameter); (**b**) bone temperature at different depths with a predrilled hole.

**Figure 7 bioengineering-11-00592-f007:**
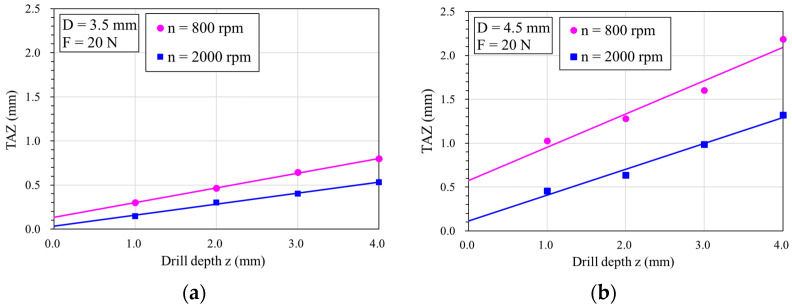
The thermally affected zone (TAZ) showed the high-risk area for thermal bone necrosis with rotational speeds of 800 rpm and 2000 rpm using two drill bit diameters: (**a**) D = 3.5 mm; (**b**) D = 4.5 mm.

**Table 1 bioengineering-11-00592-t001:** The mechanical properties of the materials used in the FE model [39,40].

Material Property	Cortical Bone	Drill Bit
Density (g/cm^3^)	1.640	7.990
Elastic modulus (MPa)	16,700	193,000
Poisson’s ratio	0.3	0.25
Yielding stress (MPa)	105	290
Ultimate stress (MPa)	106	579
Ultimate strain	0.008	0.003
Specific heat (J/kg·°C)	1640	500
Thermal conductivity (W/m·k)	0.452	16.2

## Data Availability

Data are contained within the article.

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
