# Peer review of "Assessment of Thermal Osteonecrosis during Bone Drilling Using a Three-Dimensional Finite Element Model"

_bioengineering, 2024, doi:10.3390/bioengineering11060592_

Round 1

Reviewer 1 Report

Comments and Suggestions for Authors

Review of “Assessment of thermal osteonecrosis during bone drilling using a three-dimensional finite element model” by Yung-Chuan Chen, Yi-Jung Tsai, Hao-Yuan Hsiao, Yen-Wei Chiu, You-Yao Hong, Yuan-Kun Tu and Chih-Kun Hsiao

The article studies the thermal effects of bone drilling, an important procedure in orthopedic surgery. The study employs a three-dimensional dynamic elastoplastic FE model to simulate the propagation and distribution of heat during bone drilling. The parameters influencing bone temperature, such as drill diameter, rotational speed, feed force, and predrilled holes, are examined via an approach combining experiments and FE simulations.

Major revisions are necessary before the acceptance of this manuscript. The primary issues are outlined below:

1.       The manuscript could benefit from clearer organization and structure. For instance, the methodology section could be detailed more systematically to enhance readability. Detailed thermal boundary conditions, such as initial temperatures and heat transfer coefficients, are not provided. It is essential for setting up a realistic simulation environment.

2.       All assumptions regarding the FE model should be clearly justified with adequate references, e.g. lines 155-157: “We assumed that the mechanical properties of the bone remain constant regardless of temperature while the drilling process is considered insulated (adiabatic environment). This implies that heat convection and environmental radiation impacts are not considered.”

3.       Convective and radiative heat losses are mentioned but not thoroughly modeled. The manuscript should include the equations used for modeling heat conduction and specify the thermal properties involved.

4.       The mechanisms of heat generation due to friction and plastic deformation are not adequately described. Thermal properties such as thermal conductivity, specific heat capacity, and thermal expansion coefficients are also not detailed.

5.       The manuscript’s lack of detail in the constitutive model and thermal coupling significantly weakens the study overall contribution. The latter being the pre-requisite of relevant FE predictions, it is strongly recommended to enhance the manuscript with detailed mathematical formulations. For example, the yield criterion and flow rule used to describe the elastoplastic behavior of bone should be provided along with the failure criterion If any.

6.       There is a lack of detailed information on how the constitutive model was identified and then verified against experimental data.

7.       The manuscript should provide more detail on how key variables (drill speed, feed rate, drill diameter, etc.) were controlled during the experiments. The environmental conditions under which the experiments were conducted (e.g., room temperature, humidity) should be detailed.

8.       The number of specimens used or the number of repetitions performed for each set of conditions should be given. There is no error bar in the reported data. The manuscript does not adequately discuss the limitations of the experimental setup. Potential sources of error, such as variations in bone properties or measurement inaccuracies, should be acknowledged and their impact on the results discussed.

9.       The method of temperature measurement during drilling is not sufficiently detailed. Information on the type of thermocouples or infrared cameras used, their placement, and their accuracy is needed. Additionally, how these measurements were synchronized with the drilling process should be explained.

10.   While the article references relevant studies, a more extensive literature review would strengthen the context. Including more recent research and contrasting findings could provide a deeper insight into the study novelty and significance.

Reviewer 2 Report

Comments and Suggestions for Authors

This paper is interesting and well written. The problem of the influence of drilling speed, feed rate and feed force on the maximum bone temperature generated at the drilling site during bone drilling is important and has not been well elucidated to date.

The dynamic elastoplastic finite element method was used to analyse the influence of different factors and the experimental in vitro results on porcine bone tissue and calculations are presented.  

The advantage of finite element analysis (FEA) is that the temperature can be calculated at any location alongside the assessment of the gradient and range of temperature distribution within the bone.This capability facilitates assessment of the high-risk area for thermal necrosis due to temperature elevation.

The optimal drilling speed, feed rate and feed force were estimated and additionally a two-step scheme based on pre-drilling a smaller hole prior to bone drilling was shown to be safe on the in vitro and computational models. 

The limitations of the method from the point of view of human bone tissue were also listed.

I could recommend that this paper be published after the minor revision.

Page 2, line 96: "During the bone drilling process, thebone temperature changed over time". The space in "thebone" was missing. 

Visualisation of the in vitro boreholes is recommended to support the results obtained. 

The reason for excluding the heating during the first stage of the two-stage drilling process is not clear. The authors postulated that the first state could potentially couple with the second stage, but did not present any measurement results.

Round 2

Reviewer 1 Report

Comments and Suggestions for Authors

Based on the authors' comprehensive response to the feedback and their successful incorporation of the suggested revisions, I recommend this article for publication.